# Network Pharmacology Reveals *Curcuma aeruginosa* Roxb. Regulates MAPK and HIF-1 Pathways to Treat Androgenetic Alopecia

**DOI:** 10.3390/biology13070497

**Published:** 2024-07-04

**Authors:** Aaron Marbyn L. Sintos, Heherson S. Cabrera

**Affiliations:** 1School of Chemical, Biological, and Materials Engineering and Sciences, Mapúa University, Manila 1002, Philippines; amlsintos@mymail.mapua.edu.ph; 2Department of Biology, School of Health Sciences, Mapúa University, Makati 1200, Philippines

**Keywords:** androgenetic alopecia, *Curcuma aeruginosa* Roxb., network pharmacology, molecular docking, protein–protein interaction network, enrichment analyses

## Abstract

**Simple Summary:**

Androgenetic alopecia (AGA) represents the most common form of hair loss experienced by both men and women. *Curcuma aeruginosa* Roxb., a plant known for its medicinal properties, has shown promise in reversing this hair loss disorder for its hair growth effects and anti-androgenic effects. Despite its promising potential, the mechanism of action by which it acts remains unknown. As such, this study unveiled how this plant works against hair loss by identifying its bioactive compounds, the gene its targets, and the potential mechanism involved in the therapy of AGA using network pharmacology and molecular docking. The findings revealed insights into how *C. aeruginosa* can potentially prevent AGA, highlighting its potential for developing new, safe therapies for AGA, benefiting those affected by this condition.

**Abstract:**

Androgenetic alopecia (AGA) is the most prevalent hair loss disorder worldwide, driven by excessive sensitivity or response to androgen. Herbal extracts, such as *Curcuma aeruginosa* Roxb., have shown promise in AGA treatment due to their anti-androgenic activities and hair growth effects. However, the precise mechanism of action remains unclear. Hence, this study aims to elucidate the active compounds, putative targets, and underlying mechanisms of *C. aeruginosa* for the therapy of AGA using network pharmacology and molecular docking. This study identified 66 bioactive compounds from *C. aeruginosa,* targeting 59 proteins associated with AGA. Eight hub genes were identified from the protein–protein interaction network, namely, CASP3, AKT1, AR, IL6, PPARG, STAT3, HIF1A, and MAPK3. Topological analysis of components–targets network revealed trans-verbenol, myrtenal, carvone, alpha-atlantone, and isoaromandendrene epoxide as the core components with potential significance in AGA treatment. The molecular docking verified the binding affinity between the hub genes and core compounds. Moreover, the enrichment analyses showed that *C. aeruginosa* is involved in hormone response and participates in HIF-1 and MAPK pathways to treat AGA. Overall, this study contributes to understanding the potential anti-AGA mechanism of *C. aeruginosa* by highlighting its multi-component interactions with several targets involved in AGA pathogenesis.

## 1. Introduction

Androgenic alopecia (AGA) is the most prevalent hair loss disorder, causing at least 95% of all pattern hair loss cases, such as hair thinning among women and baldness in men [1]. AGA, an androgen-dependent alopecia with genetic origin, features progressive replacement of thick, long terminal hair with fine, small vellus hair. The androgens responsible for AGA are testosterone and its biologically active metabolite, dihydrotestosterone (DHT). High levels of, or hypersensitivity to, DHT miniaturizes hair follicles, causing a shortened anagen phase of the hair growth cycle and, eventually, hair loss.

Although AGA does not pose a significant health risk, it may affect an individual’s mental and social well-being, amplifying the need for treatment [2]. Currently, only two drugs are approved by the Food and Drug Administration (FDA) to treat AGA, namely, oral finasteride and topical minoxidil. Although they exhibited favorable efficacy, side effects are reported, such as sexual dysfunction from finasteride [3] and dermatitis from minoxidil [4]. Because of this, many opt for complementary and alternative medicine (CAM) interventions. One of these is the use of herbal extracts, which have been shown to stop hair loss and encourage hair growth [5].

*Curcuma aeruginosa* Roxb. is a rhizomatous plant rich in ethnomedicinal values for treating several ailments, exhibiting a wide spectrum of pharmacologic activities such as antioxidant [6], anti-cancer [7], antimicrobial [8], anti-HIV-1 [9], uterine-relaxant [10], and anti-androgenic effects. Phytochemically, *C. aeruginosa* is a rich source of sesquiterpenes, which have been shown to exhibit anti-androgenic action in vitro and in vivo by suppressing the growth of testosterone-induced human prostate cancer cells and androgen-dependent hamster flank gland model, respectively [11]. Additionally, clinical trials have demonstrated that *C. aeruginosa* is a promising, efficient component of AGA hair tonic for slowing hair loss and stimulating hair growth [12,13]. Despite exhibiting therapeutic effects on AGA, the mechanism by which *C. aeruginosa* acts remains unknown.

Therefore, the purpose of this study is to elucidate the potential mechanism of *C. aeruginosa* against AGA through network pharmacology and molecular docking. Network pharmacology has become a commonly used tool in drug research to unveil how drugs interact with their targets, pathways, and associated disorders [14]. Meanwhile, molecular docking is utilized to confirm the potential associations of compounds and target genes predicted in the network pharmacological analysis [15].

This study is limited to investigating the influence of *C. aeruginosa* on AGA using an in silico approach, as none had been reported. Initially, the bioactive compounds of the said plant were screened out and selected. The overlap between its predicted targets and AGA-linked target genes was acquired for protein–protein interaction (PPI) construction and GO and KEGG enrichment analyses. Furthermore, the components–targets–pathways network was visualized to give a general overview of the molecular mechanisms of *C. aeruginosa* against AGA. Finally, molecular docking was employed to verify the affinity between the components and targets.

## 2. Materials and Methods

### 2.1. Screening of Bioactive Compounds in C. aeruginosa

The Indian Medicinal Plants, Phytochemistry and Therapeutics 2.0 (IMPPAT 2.0) online database (https://cb.imsc.res.in/imppat/home (accessed on 31 January 2024) [16] was used in search of compounds present in *C. aeruginosa*. This database integrates the phytochemicals of Indian medicinal plants and their pharmacokinetic properties, including drug-likeness, blood–brain barrier permeation, gastrointestinal absorption, etc. Literature mining was also conducted for further collection. The physicochemical, drug-like, and pharmacokinetic parameters of the additional compounds were computed through SwissADME (http://www.swissadme.ch/ (accessed on 1 February 2024)) [17]. The compounds that met at least three drug-likeness principles and had high gastrointestinal absorption and an oral bioavailability greater than 30% were selected and classified as bioactive ones.

### 2.2. Target Gene Prediction in C. aeruginosa

The targets of the screened potentially bioactive compounds were determined through SwissTargetPrediction (http://www.swisstargetprediction.ch/ (accessed on 30 May 2024) [18], a web server that uses a combination of two-dimensional and three-dimensional similarity metrics with known ligands to reliably predict the protein targets of compounds. For the target genes prediction *C. aeruginosa*, the canonical SMILES of compounds were copied to SwissTargetPrediction. “*Homo sapiens*” was selected as the study species to restrict the analysis of the target genes to those within humans, ensuring the relevance to human biology.

### 2.3. Target Gene Prediction in Androgenetic Alopecia

The GenCLip 3 (http://ci.smu.edu.cn/genclip3/analysis.php accessed on 30 May 2024) [19], DisGenNet (https://www.disgenet.org/ accessed on 30 May 2024) [20], Online Mendelian Inheritance in Man (OMIM) (https://omim.org/ accessed on 30 May 2024) [21], and GeneCards (https://www.genecards.org/ accessed on 30 May 2024) [22] databases were utilized to find human AGA-related targets using “androgenetic alopecia” as the keyword. The search results retrieved from the mentioned databases were combined, and duplicate targets were removed to acquire the potential target genes for treating AGA.

### 2.4. Protein–Protein Interaction Network

The target genes for the protein–protein interaction (PPI) network were obtained by determining the common targets of *C. aeruginosa* and AGA through a Venn diagram plotted in Venny 2.1 (https://bioinfogp.cnb.csic.es/tools/venny/ accessed on 30 May 2024). The PPI network of common targets was constructed by the Search Tool for Retrieval of Interacting Genes (STRING) (https://string-db.org/ accessed on 30 May 2024) [23], a database of predicted and known protein interactions, to investigate the relationship between the imported genes. The analysis was restricted to proteins found in “*Homo sapiens*” species. Afterward, the PPI network was imported into the Cytoscape (version 3.10.0) software for further analysis. CytoHubba, a Cytoscape plugin, was used to identify the potential hub target genes in the network. Three algorithms were employed—maximal clique centrality (MCC), maximum neighborhood component (MNC), and degree—to rank the best 10 genes based on the scores. The results were then intersected in which the overlapped genes represent the final set of hub targets. Moreover, MCODE was used for the cluster analysis of the PPI network with parameters set to default.

### 2.5. Gene Ontology and Pathway Enrichment Analysis

The gene ontology (GO) and Kyoto Encyclopedia of Genes and Genomes (KEGG) pathway enrichment analysis was performed to explore the biological mechanism underlying *C. aeruginosa* and its influence on AGA. GO is widely recognized for defining and describing genes from three aspects—biological process (BP), molecular function (MF), and cellular component (CC). KEGG, on the other hand, is a massive compilation of databases on drugs, genomes, enzymes, and biological pathways, among others. The enrichment analysis for the common targets was conducted using Metascape (https://metascape.org/gp/index.html accessed on 30 May 2024) [24], a gene annotation and analysis repository. Herein, the analysis was considered for “*Homo sapiens*” only, and *p*-value < 0.01 served as the cutoff. The results were taken as the top 20 based on descending −log10(*p*-value). The results were visualized in the form of a bar chart with the help of the Scientific and Research plot tool (SRplot) (http://www.bioinformatics.com.cn/SRplot accessed on 30 May 2024).

### 2.6. Components–Targets–Pathways Network Construction

Cytoscape (version 3.10.0) was used to plot a network for the top 20 KEGG pathways with the corresponding potentially bioactive compounds and targets associated with them to effectively display the interactions between *C. aeruginosa* and AGA and ultimately characterize the therapeutic mechanisms of the former on the latter.

### 2.7. Molecular Docking

Molecular docking was applied to validate the binding between the core compounds of *C. aeruginosa* and the identified hub target genes. The crystal structures of human hub genes were downloaded from the RCSB Protein Data Bank (PDB) (https://www.rcsb.org/ accessed on 30 May 2024), while the 3D structures of the *C. aeruginosa* core compounds were retrieved from the PubChem database (https://pubchem.ncbi.nlm.nih.gov/ accessed on 30 May 2024). The docking was performed using CB-Dock 2 (http://clab.labshare.co.uk/cb-dock/php/index.php accessed on 30 May 2024) [25], a protein–ligand blind docking tool that integrates the well-known molecular docking software Autodock Vina 1.1.2 to identify the binding site and predict binding pose.

## 3. Results

### 3.1. Bioactive Compounds in C. aeruginosa

Through IMPPAT, 93 compounds were found in the *C. aeruginosa* plant. Additionally, a literature search reveals a review article that described the phytochemical composition of *C. aeruginosa* [26], extracting 10 more compounds after eliminating the duplicates and those lacking structural information. In total, 103 compounds were screened, as shown in Table 1. However, only 66 were found to be bioactive after filtering with the parameters of an oral bioavailability greater than 30%, high gastrointestinal absorption, and adherence to three druglike rule-based filters, with one being the Lipinski rules. Interestingly, all compounds had bioavailability scores of 0.55, besides their high gastrointestinal absorption. A bioavailability score greater than 0.55 is assigned to any compound complying with Lipinski’s rules and is considered ideal as it indicates the compound’s optimal absorption [27].

### 3.2. Predicted Target Genes of C. aeruginosa and AGA

The 66 druglike compounds of *C. aeruginosa* had their target genes predicted through SwissTargetPrediction, yielding 795 human proteins after removing the duplicates (Appendix A). On the other hand, a total of 672 genes related to AGA were collected, of which 100 genes were from GenClip, 112 from DisGeNet, 452 from GeneCard, and 8 from OMIM. After the deletion of duplicates, 562 potential AGA-related targets were obtained (Appendix A).

### 3.3. Common Targets of C. aeruginosa and AGA

The intersection of potential *C. aeruginosa* target genes and AGA-related genes revealed 59 overlapping genes, as presented by the Venn diagram shown in Figure 1. The overlapped genes, as listed in Table A1 (see Appendix B), indicate the potential targets of *C. aeruginosa* for the therapy of AGA. They were collected for further mechanisms study of the former against the latter.

### 3.4. Protein–Protein Interaction (PPI) Network

The 59 targets associated with AGA and determined as targets for *C. aeruginosa* bioactive compounds were imported to STRING to construct a PPI network and analyze the relationship between them. An original PPI network comprising 59 nodes, of which one was disconnected, and 449 edges with an average node degree of 15.2 were produced, as shown in Figure 2. Also, it had an average local clustering coefficient of 0.629, indicating how connected the nodes were. The expected number of edges was 178, which was much lesser than the actual edges of the network, and the PPI enrichment *p*-value was observed to be <1.0 × 10^−16^. Hence, the network had significantly greater interactions than expected for the random network of similar size, and the proteins, represented by nodes, were at least partially biologically connected as a cluster.

Subsequently, the original PPI network from STRING was reconstructed in Cytoscape for visualization and further analysis. The reconstructed network, as shown in Figure 3, contained 58 nodes and 449 edges while removing one unconnected gene. The color of the node differed depending on its degree value, with darker colors reflecting greater values. The larger the degree, the more the involvement of biological functions, suggesting protein’s vital role in the network. Thus, the nodes with darker colors may serve as essential targets for the therapeutic effects of *C. aeruginosa* in AGA.

### 3.5. Identification of Hub Genes

To determine the hub targets, the CytoHubba plug-in of Cytoscape 3.10.2 was used to analyze each node by incorporating three topological algorithms, namely, maximal clique centrality (MCC), maximum neighborhood component (MNC), and degree. Figure 4 shows the top 10 genes predicted by each algorithm based on their scores. Eight genes (CASP3, AKT1, AR, IL6, PPARG, STAT3, HIF1A, and MAPK3) were screened out after intersecting the three results, as revealed in Figure 4D. These highly connected genes, as shown in Figure 4E, represent the hub target genes of *C. aeruginosa* in AGA treatment.

### 3.6. Cluster Analysis of PPI Network

MCODE plug-in was employed for cluster analysis of the PPI, yielding three cluster modules, as shown in Figure 5. Module 1 possessed 23 nodes and 186 edges; Module 2 comprised 6 nodes and 11 edges; and Module 3 comprised 3 nodes and 3 edges. Module 1 had the highest average score of 16.91, followed by Module 2 and Module 3, whose scores were 4.40 and 3.00, respectively. In the PPI network, modules with greater average scores may have more significant roles. Hence, Module 1 was the most important. Consequently, all identified hub genes were clustered in this module.

### 3.7. GO and KEGG Enrichment Analyses

GO and KEGG enrichment analyses of the 59 common targets in Metascape yielded 1055 biological processes, 64 molecular functions, 36 cellular components, and 150 KEGG pathway terms, wherein the top 20 terms in each category are visualized in Figure 6 and listed in Table A2, Table A3 and Table A4 (see Appendix B). Based on biological processes, the function of the bioactive compounds is mostly concentrated on response to peptides, lipids, and hormones, among others, suggesting that *C. aeruginosa* can modulate the hormones participating in AGA. Additionally, the majority of the genes are coded for protein in the receptor complex, transcription regulator complex, and plasma membrane protein complex, suggesting that *C. aeruginosa* targets protein complexes. Also, the cellular components included the cell body, which may indicate the potential interactions of *C. aeruginosa* bioactive compounds with cells relevant to the hair growth cycle. Molecularly, the functions of *C. aeruginosa* were mainly enriched in the activity of and binding to protein kinase and transcription factors, as well as hormone binding and nuclear receptor activity. This indicated that *C. aeruginosa* affects these proteins, which are the categories of the identified AGA-related targets of *C. aeruginosa.* Lastly, The KEGG enrichment showed the potential signaling pathways by which *C. aeruginosa* played an anti-AGA role, including MAPK and HIF-1 signaling pathways.

### 3.8. Components–Targets–Pathways Network

A network representation of the interaction between *C. aeruginosa* bioactive compounds, its potential target genes linked to AGA, and pathways associated with the targets was constructed through Cytoscape and portrayed as the components–targets–pathways network shown in Figure 7. The compounds, targets, and pathways were represented by yellow elliptical nodes, blue round rectangular nodes, and red arrow-shaped nodes, respectively. The network contained 144 nodes (66 compounds, 58 target genes, and 20 pathways) and 847 edges, showing the intricate relationships between them. Topological analysis of the components–targets network revealed trans-verbenol, myrtenal, carvone, alpha-atlantone, and isoaromandendrene epoxide as core components, exhibiting degrees of 16, 15, 15, 14, and 14, respectively, which were the highest among the compounds. Generally, this network revealed the multi-components of *C. aeruginosa* exerting synergistic multi-targeted effects against AGA.

### 3.9. Molecular Docking

The hub target genes (CASP3, AKT1, AR, IL6, PPARG, STAT3, HIF1A, and MAPK3) were molecularly docked with the core components to ascertain whether the core components (trans-verbenol, myrtenal, carvone, alpha-atlantone, and isoaromandendrene epoxide) of *C. aeruginosa* could bind to the protein targets predicted by SwissTargetPrediction. Two positive controls were also docked, namely, finasteride and minoxidil. Generally, the binding energy between the ligand compounds and receptor proteins dictates their structural stability. The lower the binding energy, the greater the affinity between the protein target and component. Based on the molecular docking studies, as shown in Figure 8, the binding energy between the core compounds and hub genes ranged from −4.6 to −8.9 kcal/mol. The results showed that the binding energies were less than 0, suggesting they spontaneously bound to one another. Binding energies less than −5 kcal/mol are believed to create more stable structures as opposed to those with greater binding energies. Hence, all molecules could bind stably to the target genes with strong affinity except for carvone and myrtenal when bound to HIF1A. AKT1 and AR, among the target genes, had the highest binding affinity with the core compounds. This may imply that targeting these proteins plays an essential role in AGA treatment by *C. aeruginosa* bioactive compounds. Strikingly, alpha-atlantone and isoaromandendrene epoxide demonstrated superior affinity to AR when compared to controls. The top four stable ligand-receptor complexes were selected for visualization, as shown in Figure 9, and their binding sites were tabulated in Table 2. Since both alpha-atlantone and isoaromandendrene epoxide are sesquiterpenes, they are bound to similar amino acid residue sites in the AKT1 complex.

## 4. Discussion

AGA is a multifactorial hair loss disorder, and those suffering from this disease have limited options for medical treatment. Drugs like finasteride and minoxidil cause side effects, restricting their long-term administration. Moreover, invasive treatments like hair transplantation and platelet-rich plasma (PRP) require repeated procedures, resulting in costly investments [28,29]. Currently, topical herbal preparations are becoming more commonly available due to their greater compliance rate, broader active spectrum, more affordable price, and lesser side effects [30,31]. Thus, they are anticipated to be extensively utilized for AGA complementary and alternative medicine. *C. aeruginosa*, as a topical preparation, has been shown to exhibit anti-androgenic and hair growth effects due to its phytochemical content that may target different pathways involved in AGA. Accordingly, network pharmacology fits as a valid method for elucidating its multicomponent–multitarget anti-AGA mechanism.

The network pharmacology results revealed five compounds—trans-verbenol, myrtenal, carvone, alpha-atlantone, and isoaromandendrene epoxide—that might be the core components in *C. aeruginosa* and enable it to induce therapeutic effects against AGA. However, no studies reported their anti-androgenic and trichogenic effects yet, recommending further in vivo and in vitro studies to test their anti-AGA potential. Molecular docking analysis confirmed that the core components bound with the eight hub genes—CASP3, AKT1, AR, IL6, PPARG, STAT3, HIF1A, and MAPK3—implying potential interactions and modulation of these genes to treat AGA.

CASP3, whose role is central in executing cell apoptosis, is overexpressed in the bald area of AGA patients in the early courses of the disease, revealing the presence of inflammation and apoptosis at such stages [32]. Activation of CASP3 inhibits the PI3K/AKT signaling pathway, which mediates extracellular signals and intracellular responses and is critically involved in the regulation of cell proliferation, growth, and differentiation. This, in turn, stops hair follicles from transitioning from the resting phase to the anagen phase, blocks cell proliferation, promotes apoptosis, and finally degenerates hair follicles [33]. As such, inhibiting CASP3 activation helps reverse hair follicles’ entry to the abnormal and degenerative anagen phases and stimulates hair growth.

Likewise, AKT1, one of the relevant serine/threonine protein kinases, regulates cell apoptosis and proliferation and serves as the main downstream molecule of the PI3K/AKT pathway, whose role is necessary for de novo hair follicle regeneration [34]. In response to extracellular signals, AKT1 can either act as a positive or negative regulator of the PI3K/AKT pathway via PIK3, leading to AR expression regulation [35].

AR-bound DHT is the main cause of AGA, with DHT-AR signaling strongly associated with AGA pathogenesis [36,37,38]. AR is primarily expressed in hair follicles, particularly dermal papilla cells (DPCs) [39]. When DHT binds to AGA, expression of growth inhibition factors (i.e., DKK-1, TGF-β, and IL-6) is triggered [40,41]. IL-6 suppresses hair shaft elongation by inhibiting matrix cell proliferation and, thereby, stimulates hair follicle regression [42]. Additionally, PPARG also discourages hair growth, but by promoting mitochondrial activity [43]. Essentially, DHT-AR signaling facilitates the miniaturization of hair follicles, resulting in the apoptosis of keratinocytes [44] and DPCs [45] and, eventually, AGA progression. Therefore, treatment for AGA may benefit most from inhibiting AR expression due to the major role of AR in AGA.

Contrary to the alopecia-inducing effects of the earlier hub genes, STAT3, HIF1A, and MAPK3 are reported to counteract hair loss. STAT3 is required in the hair cycle during the onset of anagen because it activates keratinocytes for the continuation of the hair cycle [46]. Loss of STAT3 functions in keratinocytes increases apoptotic hair follicle stem cells (HFSCs), impairing the hair cycle process [47]. Contrastingly, gain of function raises progenitor cells and HFSCs above the bulge region, ensuring proper maintenance and growth of hair follicles [48]. With this, STAT3 regulation is critical in maintaining hair cycling and growth. Meanwhile, HIF1A regulates trichogenic gene expression in DPCs [49], suggesting a similar function to minoxidil, which exerts trichogenic effects ascribed by its vasodilating properties [50]. Lastly, MAPK’s role in regenerating HFSCs, inducing anagen hair cycle, and modulating root hair tip growth is important in hair growth stimulation [51,52]. Specifically, when MAPK3 was upregulated, hair growth improved [53].

The GO enrichment analysis suggested that *C. aeruginosa* may regulate hormones, such as androgen, estrogen, and cortisol. Androgens, like testosterone and DHT, activate AR signaling, upregulating genes involved in hair growth suppression resulting from growth inhibition factors, vascular regression around dermal papilla [54], apoptosis [55], and DPC aging [56]. On the other hand, estrogen maintains hair follicle cycling [57], encourages healthy hair growth by activating the Wnt/β-catenin signaling pathway to sustain HFSC differentiation and proliferation [58,59], and shields hair follicles from oxidative stress and eventually hair follicle aging by modulating antioxidant enzymes [60]. Lastly, corticosterone, the cortisol counterpart in mice and the main stress hormone, disallows the entry of HFSCs into the anagen phase [61]. Thus, chronic stress can quicken hair aging and loss by influencing HFSC.

On the other hand, the KEGG enrichment showed the potential signaling pathways by which *C. aeruginosa* played an anti-AGA role, including MAPK and HIF-1 signaling pathways. The identified hub genes were implicated in the MAPK pathway (AKT1, CASP3, and MAPK3) and the HIF-1 pathway (AKT1, GIF1A, IL6, MAPK3, and STAT3) through which core compounds may act to modulate them. The combined effects of the compounds through their regulation of MAPK and HIF-1 pathways, together with direct interaction with other hub genes, create a synergistic approach to address various aspects of AGA pathogenesis. MAPK pathway is important in regulating normal cell survival, migration, proliferation, and migration [62]. In hair, MAPK has been revealed to amplify growth factor production [63], regulate the hair cycle and quiescence of HFSC [51], and promote HFSC differentiation and proliferation [64], thereby influencing hair follicle morphogenesis and regeneration. Currently, four MAPK signal transduction pathways are known in mammalian cells: extracellular signal-regulated kinases (ERKs) which stimulate DPC proliferation and anagen phase [65,66]; and p38 MAPKs and Jun N-terminal kinases (JNKs), both of which control Wnt/β-catenin pathway [67,68], the master regulator of hair cells.

Whereas the HIF-1 pathway has been shown to govern hair regeneration, regulating the size and shape of dermal papilla [69,70]. AGA has been associated with insufficient nutrient supply and reduced blood vessels. Consequently, HIF stimulation can come into play in this by regulating neovascularization and regeneration as DPCs react to hypoxia [71]. The HIF-1 pathway is strongly linked to the mechanism of action of minoxidil attributed to its vasodilating properties [72]. Clinical trials showed that the combination of minoxidil and *C. aeruginosa* stimulated hair growth more effectively than minoxidil alone [12,13]. The topical application of *C. aeruginosa* complimented minoxidil as it increased hair growth and decreased hair shedding by enhancing penetration [12,13]. This may be caused by *C. aeruginosa*’s potential regulation of the HIF-1 pathway, which is said to increase vasodilation, promoting conducive conditions for hair growth. Figure 10 shows the potential mechanism of *C. aeruginosa* against AGA. It is essential to consider that although MAPK and HIF-1 pathways influence AGA, their roles are part of a complex cascade of events rather than a direct one.

Notably, Module 3 derived from the cluster analysis of the PPI network contained SRD5A2, SRD5A1, and CYP17A1, which are involved in androgen biosynthesis. SRD5As amplify DHT production in hair follicles of the scalp, causing AGA [73]. To date, steroidal drugs finasteride and dutasteride are used to treat AGA by acting as SRD5A inhibitors. Finasteride selectively inhibits SRD5A2 while dutasteride inhibits both SRD5A1 and SRD5A2. Accordingly, a prior study suggested that the potential mechanism of C. aeruginosa for its anti-androgenic effect is SRD5A inhibition [11]. Its sesquiterpene content, specifically germacrone, was on par with finasteride when it came to exhibiting anti-androgenic activity. It inhibited SRD5A to a similar extent as finasteride in suppressing the growth of testosterone-induced growth of human prostate cancer cells (in vitro) and hamster flank gland model (in vivo), thereby suggesting *C. aeruginosa* as a novel SRD5A inhibitor. Similarly, CYP17A1 is needed in DHT production, both in anterior and posterior routes [74,75]. Minoxidil suppresses CYP17A1 to inhibit AGA [76]. Therefore, synergizing this with MAPK and HIF-1 pathways may influence both anti-androgenic activities and hair growth effects, promoting healthy hair follicles and eventually preventing AGA.

While AGA affects both men and women, men are more commonly and severely impacted because of innate higher levels of androgens. Although *C. aeruginosa* exhibited trichogenic activities that could impact women too, men with genetic predispositions to AGA are likely to see more significant benefits from *C. aeruginosa* given its anti-androgenic effects. Additionally, younger men, whose AGA symptoms are not yet more pronounced, may experience greater advantages from early intervention using it to potentially prevent AGA. Further research is recommended to fully comprehend these effects across different patient populations. In terms of safety, no side effects were reported on the topical application of *C. aeruginosa* in clinical trials [12,13], as opposed to those associated with finasteride and minoxidil, namely, sexual dysfunction and dermatological problems. This underscores the potential of *C. aeruginosa* as an AGA treatment, warranting further investigation.

## 5. Conclusions

In summary, this study combined network pharmacology and molecular docking to elucidate the active compounds, putative targets, and potential mechanisms of *C. aeruginosa* in the treatment of AGA. The results pinpointed that bioactive compounds in *C. aeruginosa*, such as trans-verbenol, myrtenal, carvone, alpha-atlantone, and isoaromandendrene epoxide, may play a crucial role in AGA by eliciting their effects on key target genes, including CASP3, AKT1, AR, IL6, PPARG, STAT3, HIF1A, and MAPK3. The molecular docking studies indicated that these bioactive components of *C. aeruginosa* could effectively act on these targets. Also, enrichment analyses revealed that *C. aeruginosa* may play its therapeutic role against AGA by modulating both HIF-1 and MAPK pathways, offering new approaches in the treatment and prevention of AGA. Substantially, the findings showed that *C. aeruginosa* could treat AGA via a mechanism involving multiple components, targets, and pathways. Hence, the valuable results may imply that *C. aeruginosa* could be a promising option in developing new drugs for AGA. As bound by the limitations of bioinformatics data and in silico network pharmacology and molecular docking analysis, however, experimental exploration and further confirmation via in vivo and in vitro studies are recommended to verify the findings of this study.

## Figures and Tables

**Figure 1 biology-13-00497-f001:**
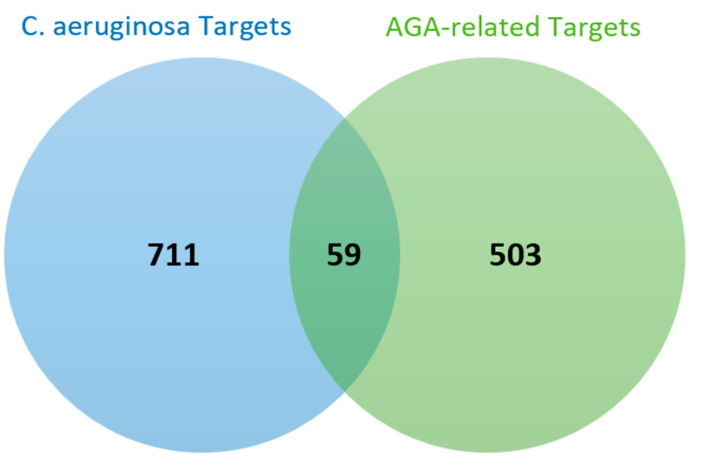
The 59 overlapping targets between *C. aeruginosa* and AGA identified by Venn diagram.

**Figure 2 biology-13-00497-f002:**
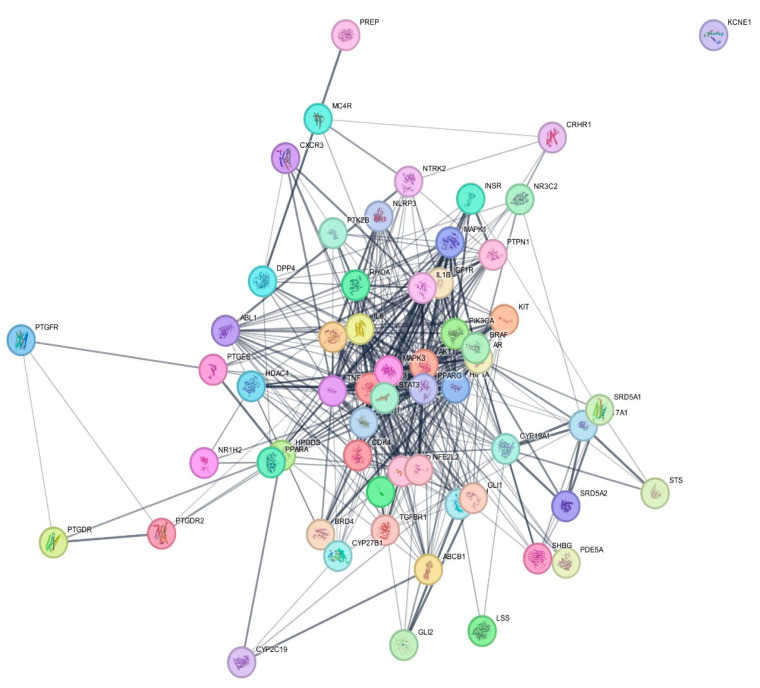
Original PPI network of 59 potential targets of *C. aeruginosa* in AGA constructed by STRING.

**Figure 3 biology-13-00497-f003:**
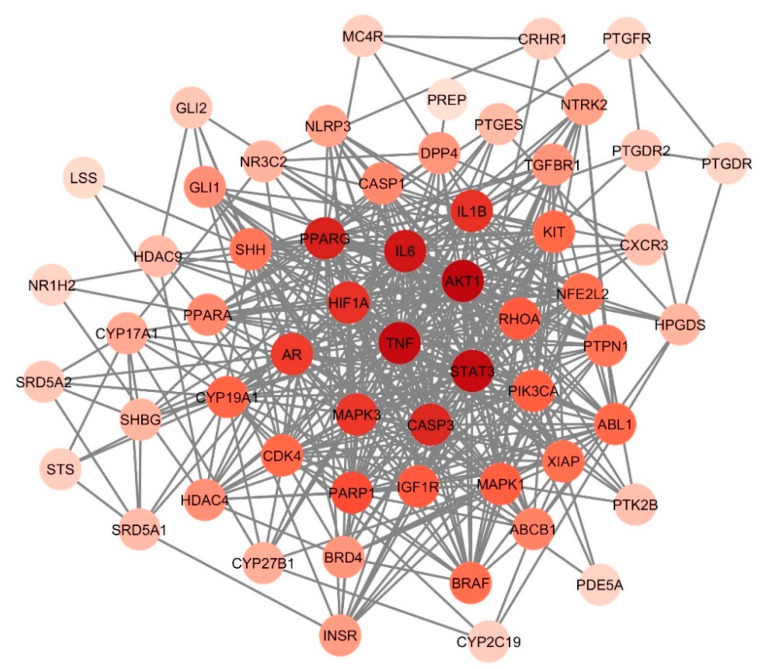
The PPI network (58 nodes and 449 edges) showing the degree of the targets reconstructed by Cytoscape 3.10.2. The darker the color of the node, the greater its degree.

**Figure 4 biology-13-00497-f004:**
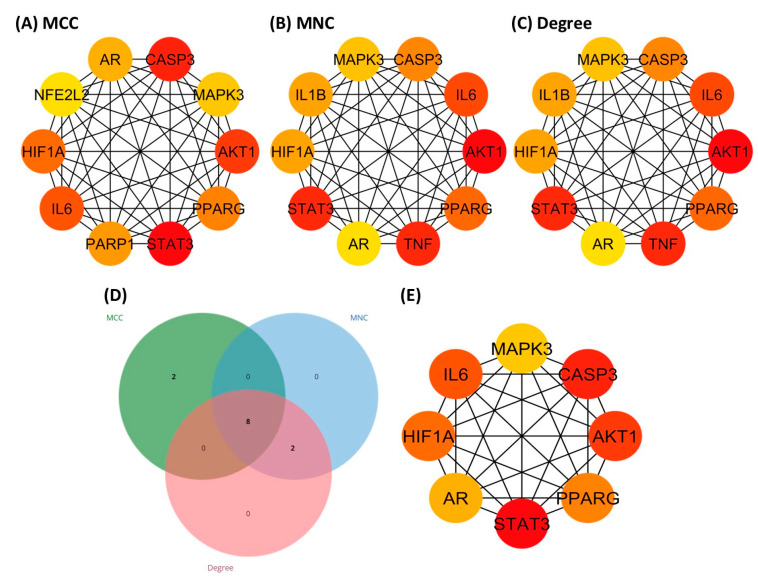
The top 10 hub gene networks of *C. aeruginosa* bioactive compounds against AGA from the employment of (**A**) maximal clique centrality, (**B**) maximum neighborhood component (MNC), and (**C**) degree. The warmer the color, the higher the score. The score correlates with the rank of genes in the network. (**D**) The Venn diagram intersecting the results of the three algorithms, revealing eight hub genes. (**E**) The PPI network of eight hub genes. Node color denotes interaction degree (red for high degree, orange for intermediate degree, and yellow for low degree).

**Figure 5 biology-13-00497-f005:**
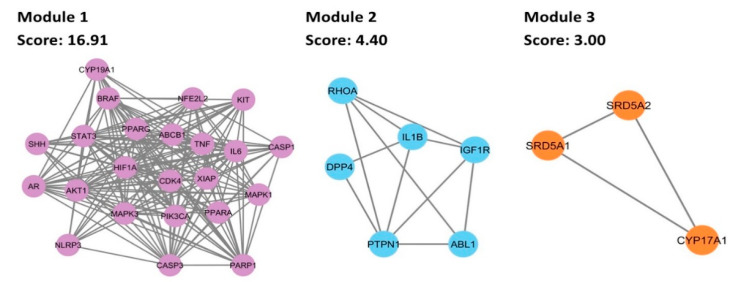
The modules obtained from the cluster analysis of the PPI network.

**Figure 6 biology-13-00497-f006:**
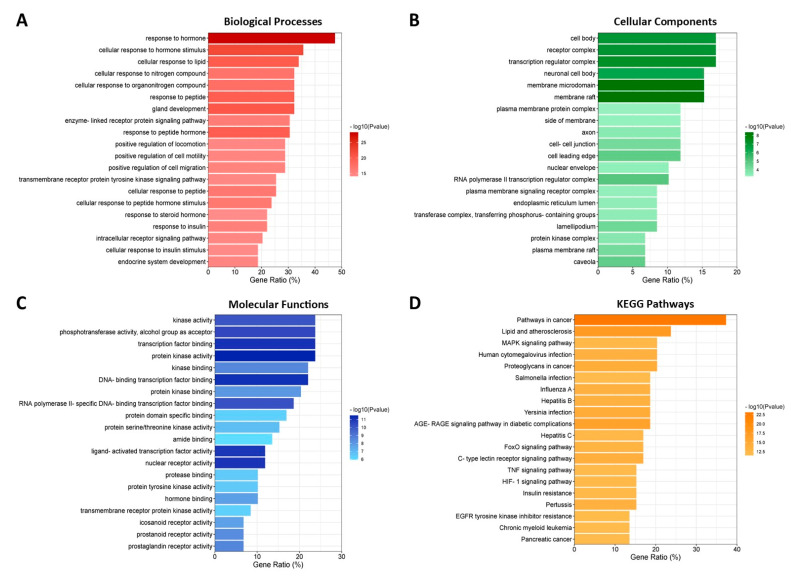
Enrichment analyses of *C. aeruginosa* potential targets in AGA for the top 20 GO annotations and KEGG pathways: (**A**) GO biological processes, (**B**) GO cellular components, (**C**) GO molecular functions, and (**D**) KEGG pathways.

**Figure 7 biology-13-00497-f007:**
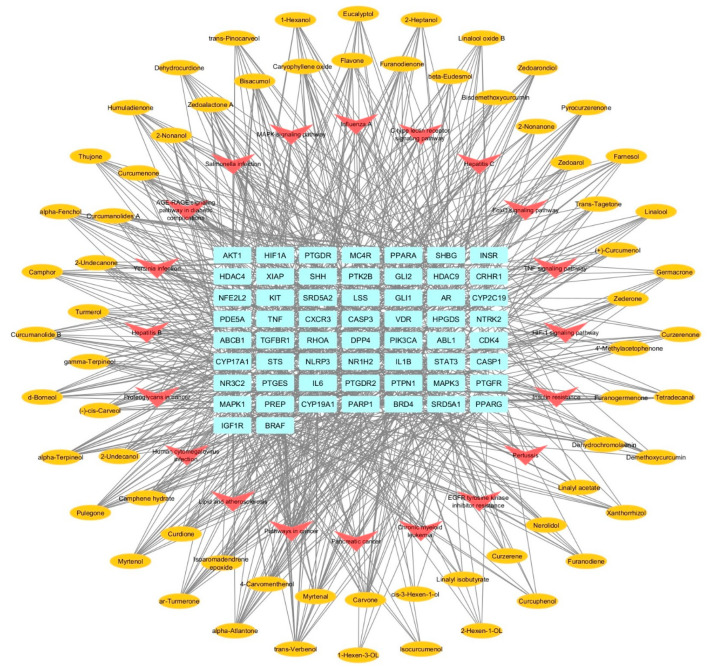
The components–targets–pathways network displaying the potential mechanism of *C. aeruginosa* against AGA (yellow ellipses: compounds; red arrows: pathways; blue rectangles: targets).

**Figure 8 biology-13-00497-f008:**
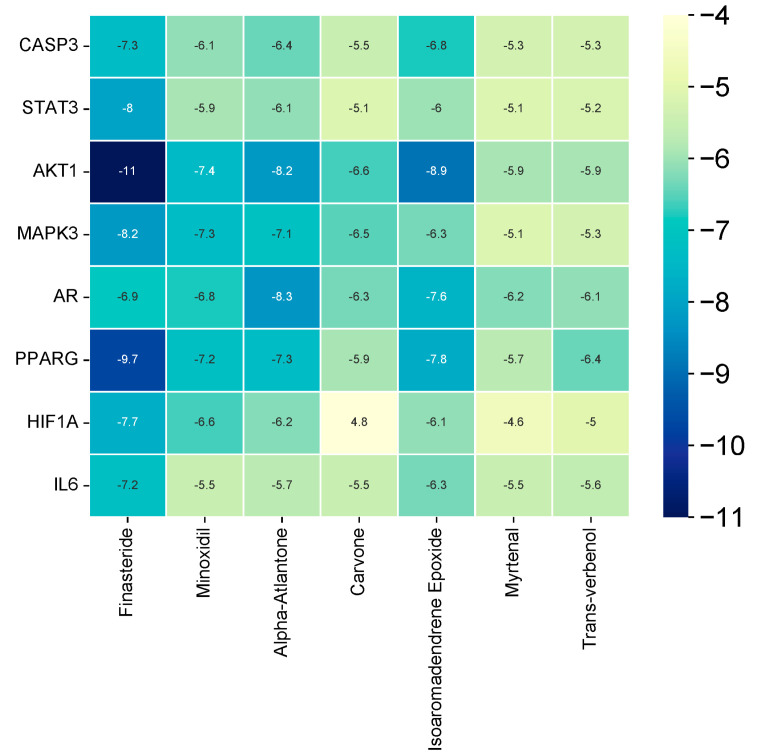
Heatmap of the molecular docking of *C. aeruginosa* core compounds with hub target genes. The bluer the color, the greater the binding energy and binding affinity between the ligand and the receptor.

**Figure 9 biology-13-00497-f009:**
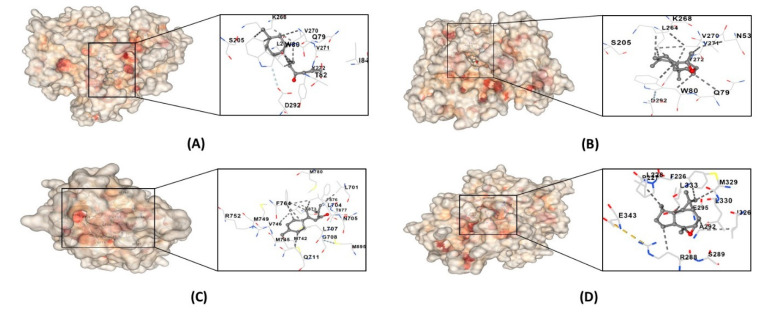
The top four ligand-receptor complexes: (**A**) AKT1-alpha-atlantone, (**B**) AKT1-isoaromandendrene epoxide, (**C**) AR-alpha-atlantone, and (**D**) PPARG- isoaromandendrene epoxide.

**Figure 10 biology-13-00497-f010:**
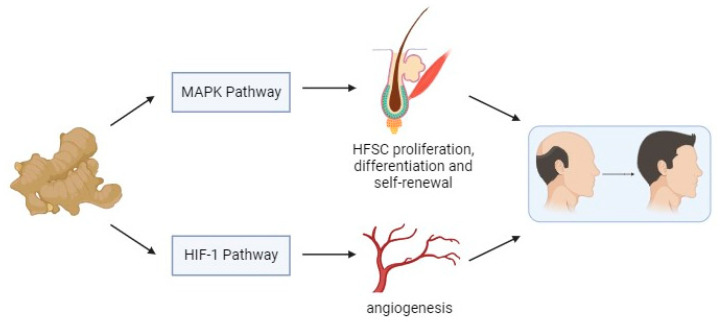
The potential mechanism of *Curcuma aeruginosa* Roxb. against androgenetic alopecia. Collectively, in the treatment of AGA, the MAPK pathway promotes hair follicle proliferation, differentiation, and self-renewal to maintain the hair cycle, and the HIF-1 pathway improves hair vascularization and nutrient supply conducive to hair growth.

**Table 1 biology-13-00497-t001:** Bioactive compounds in *C. aeruginosa*.

No.	Compound	MW	OB	GIA	Drug-Likeness				
					Lipinski	Ghose	Veber	Muegge	Egan
1	Zedoarol	246.31	0.55	High	Yes	Yes	Yes	Yes	Yes
2	Myrcene *	136.24	0.55	Low	Yes	Yes	No	Yes	Yes
3	Trans-Tagetone	152.24	0.55	High	Yes	Yes	No	Yes	Yes
4	Furanodienone	230.31	0.55	High	Yes	Yes	Yes	Yes	Yes
5	gamma-Terpinene *	136.24	0.55	Low	Yes	Yes	No	Yes	Yes
6	1-Hexen-3-OL	100.16	0.55	High	Yes	Yes	No	Yes	Yes
7	p-Cymene *	134.22	0.55	Low	Yes	Yes	No	Yes	Yes
8	Curdione	236.36	0.55	High	Yes	Yes	Yes	Yes	Yes
9	Myrtenal	150.22	0.55	High	Yes	Yes	No	Yes	Yes
10	Germacrone	218.34	0.55	High	Yes	Yes	Yes	Yes	Yes
11	Isocurcumenol	234.34	0.55	High	Yes	Yes	Yes	Yes	Yes
12	1-Hexanol	102.18	0.55	High	Yes	Yes	No	Yes	Yes
13	beta-Cubebene *	204.36	0.55	Low	Yes	Yes	Yes	Yes	Yes
14	Eucalyptol	154.25	0.55	High	Yes	Yes	No	Yes	Yes
15	beta-Elemene *	204.36	0.55	Low	No	Yes	Yes	Yes	Yes
16	Furanogermenone	232.32	0.55	High	Yes	Yes	Yes	Yes	Yes
17	Curzerene	216.32	0.55	High	Yes	Yes	Yes	Yes	Yes
18	(+)-Curcumenol	234.34	0.55	High	Yes	Yes	Yes	Yes	Yes
19	4-Carvomenthenol	154.25	0.55	High	Yes	Yes	No	Yes	Yes
20	(1R)-2-methyl-5-propan-2-ylbicyclo[3.1.0]hex-2-ene *	136.24	0.55	Low	Yes	Yes	No	Yes	Yes
21	Curzerenone	230.31	0.55	High	Yes	Yes	Yes	Yes	Yes
22	alpha-Selinene *	204.36	0.55	Low	No	Yes	Yes	Yes	Yes
23	cis-3-Hexen-1-ol	100.16	0.55	High	Yes	Yes	No	Yes	Yes
24	d-Borneol	154.25	0.55	High	Yes	Yes	No	Yes	Yes
25	Terpinolene *	136.24	0.55	Low	Yes	Yes	No	Yes	Yes
26	beta-Farnesene *	204.36	0.55	Low	No	Yes	Yes	Yes	Yes
27	Curcumanolide B	234.34	0.55	High	Yes	Yes	Yes	Yes	Yes
28	Curcumanolides A	234.34	0.55	High	Yes	Yes	Yes	Yes	Yes
29	2-Hexen-1-OL	100.16	0.55	High	Yes	Yes	No	Yes	Yes
30	Humulene *	204.36	0.55	Low	No	Yes	Yes	Yes	Yes
31	Pulegone	152.24	0.55	High	Yes	Yes	No	Yes	Yes
32	Thujone	152.24	0.55	High	Yes	Yes	No	Yes	Yes
33	(+)-delta-Cadinene *	204.36	0.55	Low	No	Yes	Yes	Yes	Yes
34	(-)-cis-Carveol	152.24	0.55	High	Yes	Yes	No	Yes	Yes
35	Camphor	152.24	0.55	High	Yes	Yes	No	Yes	Yes
36	Linalool	154.25	0.55	High	Yes	Yes	No	Yes	Yes
37	alpha-Pinene *	136.24	0.55	Low	Yes	Yes	No	Yes	Yes
38	Carvone	150.22	0.55	High	Yes	Yes	No	Yes	Yes
39	beta-Pinene *	136.24	0.55	Low	Yes	Yes	No	Yes	Yes
40	alpha-Fenchol	154.25	0.55	High	Yes	Yes	No	Yes	Yes
41	alpha-Terpineol	154.25	0.55	High	Yes	Yes	No	Yes	Yes
42	Sabinene *	136.24	0.55	Low	Yes	Yes	No	Yes	Yes
43	trans-Pinocarveol	152.24	0.55	High	Yes	Yes	No	Yes	Yes
44	Caryophyllene oxide	220.36	0.55	High	Yes	Yes	Yes	Yes	Yes
45	Phytol *	296.54	0.55	Low	No	Yes	No	No	No
46	(Z)-beta-Ocimene *	136.24	0.55	Low	Yes	Yes	No	Yes	Yes
47	gamma-Elemene *	204.36	0.55	Low	No	Yes	Yes	Yes	Yes
48	beta-Selinene *	204.36	0.55	Low	No	Yes	Yes	Yes	Yes
49	beta-Caryophyllene *	204.36	0.55	Low	No	Yes	Yes	Yes	Yes
50	(E)-beta-ocimene *	136.24	0.55	Low	Yes	Yes	No	Yes	Yes
51	Camphene *	136.24	0.55	Low	Yes	Yes	No	Yes	Yes
52	Limonene *	136.24	0.55	Low	Yes	Yes	No	Yes	Yes
53	trans-Verbenol	152.24	0.55	High	Yes	Yes	No	Yes	Yes
54	Allo-Aromadendrene *	204.36	0.55	Low	Yes	Yes	Yes	Yes	Yes
55	2-Heptanol	116.2	0.55	High	Yes	Yes	No	Yes	Yes
56	Myrtenol	152.24	0.55	High	Yes	Yes	No	Yes	Yes
57	beta-Bisabolene *	204.36	0.55	Low	No	Yes	Yes	Yes	Yes
58	Curcumenone	234.43	0.55	High	Yes	Yes	Yes	Yes	Yes
59	2-Undecanol	172.31	0.55	High	Yes	Yes	Yes	Yes	Yes
60	gamma-Terpineol	154.24	0.55	High	Yes	Yes	No	Yes	Yes
61	Humuladienone	220.36	0.55	High	Yes	Yes	Yes	Yes	Yes
62	(-)-beta-Curcumene *	204.36	0.55	Low	No	Yes	Yes	Yes	Yes
63	Dehydrocurdione	234.43	0.55	High	Yes	Yes	Yes	Yes	Yes
64	Tetradecanal	212.38	0.55	High	Yes	Yes	No	Yes	Yes
65	Bisacumol	218.34	0.55	High	Yes	Yes	Yes	Yes	Yes
66	Tricyclene *	136.24	0.55	Low	Yes	Yes	No	Yes	Yes
67	Linalyl acetate	196.29	0.55	High	Yes	Yes	Yes	Yes	Yes
68	4′-Methylacetophenone	134.18	0.55	High	Yes	Yes	No	Yes	Yes
69	Xanthorrhizol	218.34	0.55	High	Yes	Yes	Yes	Yes	Yes
70	1-Methyl-4-(prop-1-en-2-yl)benzene *	132.21	0.55	Low	Yes	Yes	No	Yes	Yes
71	Linalyl isobutyrate	224.34	0.55	High	Yes	Yes	Yes	Yes	Yes
72	alpha-Guaiene *	204.36	0.55	Low	No	Yes	Yes	Yes	Yes
73	2-Nonanol	144.26	0.55	High	Yes	Yes	No	Yes	Yes
74	2-Nonanone	142.24	0.55	High	Yes	Yes	No	Yes	Yes
75	3,7(11)-Eudesmadiene *	204.46	0.55	Low	No	Yes	Yes	Yes	Yes
76	Camphene hydrate	154.25	0.55	High	Yes	Yes	No	Yes	Yes
78	Curcuphenol	218.34	0.55	High	Yes	Yes	Yes	Yes	Yes
79	Turmerol	220.36	0.55	High	Yes	Yes	Yes	Yes	Yes
80	2-Undecanone	170.3	0.55	High	Yes	Yes	Yes	Yes	Yes
81	beta-Eudesmol	222.37	0.55	High	Yes	Yes	Yes	Yes	Yes
82	Farnesol	222.37	0.55	High	Yes	Yes	Yes	Yes	Yes
83	alpha-Terpinene *	136.23	0.55	Low	Yes	Yes	No	Yes	Yes
84	cis-beta-Farnesene *	204.36	0.55	Low	No	Yes	Yes	Yes	Yes
85	Zingiberene *	204.36	0.55	Low	No	Yes	Yes	Yes	Yes
86	(+)-beta-Phellandrene *	136.23	0.55	Low	Yes	Yes	No	Yes	Yes
87	alpha-Curcumene *	202.34	0.55	Low	No	Yes	Yes	Yes	Yes
88	ar-Turmerone	216.32	0.55	High	Yes	Yes	Yes	Yes	Yes
89	3-(1,5-Dimethyl-4-hexenyl)-6-methylene-1-cyclohexene *	204.36	0.55	Low	Yes	Yes	No	Yes	Yes
90	Linalool oxide B	170.25	0.55	High	Yes	Yes	Yes	Yes	Yes
91	delta-Elemene *	204.36	0.55	Low	No	Yes	Yes	Yes	Yes
92	alpha-Atlantone	218.34	0.55	High	Yes	Yes	Yes	Yes	Yes
93	alpha-Phellandrene *	136.24	0.55	Low	Yes	Yes	No	Yes	Yes
94	Nerolidol	222.37	0.55	High	Yes	Yes	Yes	Yes	Yes
95	Flavone	222.24	0.55	High	Yes	Yes	Yes	Yes	Yes
96	Zedoalactone A	266.36	0.55	High	Yes	Yes	Yes	Yes	Yes
97	Zedoarondiol	252.35	0.55	High	Yes	Yes	Yes	Yes	Yes
98	Furanodiene	216.32	0.55	High	Yes	Yes	Yes	Yes	Yes
99	Zederone	246.30	0.55	High	Yes	Yes	Yes	Yes	Yes
100	Pyrocurzerenone	212.29	0.55	High	Yes	Yes	Yes	Yes	Yes
101	Dehydrochromolaenin	210.27	0.55	High	Yes	Yes	Yes	Yes	Yes
102	Isoaromadendrene epoxide	220.35	0.55	High	Yes	Yes	Yes	Yes	Yes
103	Demethoxycurcumin	338.35	0.55	High	No	Yes	Yes	Yes	Yes

* Did not meet the screening conditions.

**Table 2 biology-13-00497-t002:** Binding site interactions of top 4 ligand–protein complexes.

Compound	Protein	Binding Sites
alpha-atlantone	AR	Leu701, Leu704, Asn705, Leu707, Gly708, Gln711, Trp741, Met742, Met745, Val746, Met749, Arg752, Phe764, Met780, Met787, Leu873, Phe876, THR877 Leu880, Phe891, Met895, Ile899
	AKT1	Asn53, Asn54, Ala58, Gln59, Leu78, Gln79, Trp80, Thr82, Ile84, Asn199, Val201, Ser205, Leu210, Thr211, Leu264, Lys268, Val270, Val271, Tyr272, Ile290, Thr291, Asp292
isoaromandendrene epoxide	AKT1	Glu17, Tyr18, Asn53, Asn54, Ser56, Ala58, Gln59, Gln79, Trp80, Thr81, Thr82, Ile84, Glu85, Arg86, Lys179, Val201, Ser205, Leu210, Thr211, Leu213, Tyr263, Leu264, Lys268, Val270, Val271, Tyr272, Arg273, Asp274, Asn279, Thr291, Asp292, Phe293, Gly294, Cys296, Lys297, Glu298, Tyr326
	PPARG	Phe226, Pro227, Leu228, Gly284, Cys285, Arg288, Ser289, Glu291, Ala292 Glu295, Ile296, Ile325, Ile326, Met329, Leu330, Leu333, Val339, Leu340, Ile341, Ser342, Glu343, Gly344, Met364

## Data Availability

The data and results presented in this study are available upon request from the first and corresponding author.

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
