# Peer review of "Network Pharmacology Reveals Curcuma aeruginosa Roxb. Regulates MAPK and HIF-1 Pathways to Treat Androgenetic Alopecia"

_biology, 2024, doi:10.3390/biology13070497_

Round 1

Reviewer 1 Report

Comments and Suggestions for Authors

The study utilized network pharmacology to identify potential bioactive compounds and mechanism of C. aeruginosa to target Androgenetic alopecia (AGA). The work follows a generic pipeline of screening some compounds based on generic filters, docking with the targets, and suggest predictions without any validation. Although the work is insilico, and I understand that there is only so much that can be done in a prediction based approach, the work needs to dig-in a little deeper to be considered for publication.

Major Comments

1) Line 80: The authors mention the Lipinski Rules as part of the filtering criteria. However, these guidelines (and not rules) do not always reflect the drug-likness of compounds. There are many drugs that do not follow the Lipinkski “Rules”. Have the authors played with the numbers or values of these conditions? How does the chemical space differ after altering the filtering criteria?

2) Line 81-83: What are the other two druglikness properties? Did the authors also look at the Toxicity in the ADME(T) filtering? If not, why?

3) Why were other databases like DrugBank, and Zinc not considered to increase the chemical space for screening?

4) The work includes a lot of predictions from different freely available prediction tools and softwares, I do not see any knowledge-base applied to the workflow.

5) Could the authors comment on why only a generic molecular docking was performed without taking into account the pharmacophor mapping?

6) Are there any existing drugs that can be repurposed for AGA?

7) Why were compound structural simlarity and fingerprinting not part of the screening process?

8) Docking in general can be very subjective. It is essential to identify a viable active site and not just a protein pocket. Docking depends heavily on the size of the frame where you want to dock. Molecules will dock to any open site that is big enough which might not be the “active” site.

9) The paper does not mention and benchmarking studies, nor any statistical analysis on the accuracy of the predictions.

The paper as a whole has the potential to be more informative and be published after a major revision and incorporation of more biological knowledge-base and not rely upon the prediction of some softwares.

Comments on the Quality of English Language

English is fine. Might need minor revisions.

Author Response

1) Line 80: The authors mention the Lipinski Rules as part of the filtering criteria. However, these guidelines (and not rules) do not always reflect the drug-likness of compounds. There are many drugs that do not follow the Lipinkski “Rules”. Have the authors played with the numbers or values of these conditions? How does the chemical space differ after altering the filtering criteria?

We have eliminated the Lipinski Rules in the filtering criteria. Instead, they should meet any of 3 of the 5 drug-based rule filters (Lipinski, Ghose, Veber, Muegge, and Egan) or more. In any case, nothing was affected in my results because all the rejected compounds were due to their low gastrointestinal absorption. I apologize for the confusion from the results in this section. We only included the screened-out bioactive compounds in the table, omitting the unfavorable ones. We have revised the table wherein we classified the compounds with asterisks for those not meeting the criteria, showing the chemical space before and after filtering.

2) Line 81-83: What are the other two druglikness properties? Did the authors also look at the Toxicity in the ADME(T) filtering? If not, why?

The other two drug-likeness properties may be Ghose, Veber, Muegge, and Egan. For the screening condition, the compounds should adhere to at least 3 of the 5 drug-likeness properties (Lipinski, Ghose, Veber, Muegge, and Egan). As for the determination of pharmacokinetic properties, we only used SwissADME, which does not include toxicity, to initially filter out the compounds with unfavorable pharmacokinetic properties and identify the bioactive compounds for further analyses.  

3) Why were other databases like DrugBank, and Zinc not considered to increase the chemical space for screening?

The aim of this study is to elucidate the potential mechanism of C. aeruginosa against AGA. Since C. aeruginosa is a plant, IMPPAT (Indian Medicinal Plants, Phytochemistry and Therapeutics) is an optimal choice for our study as it is highly curated and specific for plant-based compounds and is, thus, relevant to our main objective. Considering other databases may lower the applicability and specificity of our results.

4) The work includes a lot of predictions from different freely available prediction tools and software, I do not see any knowledge-base applied to the workflow.

While it is true that this work utilized freely available prediction tools and software, the methods used are widely accepted in drug discovery, ensuring relevant predictions. Also, the computational predictions were validated with literature, particularly the Discussion (Section 4, Lines 304-432), grounding the study.

5) Could the authors comment on why only a generic molecular docking was performed without taking into account the pharmacophore mapping?

One of the objectives of the study was to verify the binding affinities between target genes predicted by network pharmacology, and plant bioactive compounds. The exclusion of pharmacophore mapping aligns with the widely accepted network pharmacology framework because it is standard practice in network pharmacology to employ molecular docking for initial validation without pharmacophore mapping. As a study at an exploratory stage, it is not necessary to fully characterize them but rather determine potential interactions, sufficing the use of molecular docking as a standalone technique for initial screening.

6) Are there any existing drugs that can be repurposed for AGA?

Based on the literature, there are. But in relation to our study, most of the core compounds (i.e., myrtenal and carvone) have demonstrated diverse pharmacological properties such as anti-diabetic, anti-inflammatory, antioxidant, anti-cancer, antiviral, etc. Therefore, they may be potential candidates to be repurposed for AGA upon further investigation. Moreover, it is found in our study that C. aeruginosa exerts its effects against AGA by HIF-1 and MAPK pathways. Existing drugs that act on these pathways may be repurposed since HIF-1 and MAPK pathways are targets for cardiovascular disease and cancer treatments, respectively.

7) Why were compound structural similarity and fingerprinting not part of the screening process?

The primary purpose of this study is to elucidate the underlying mechanism of C. aeruginosa against AGA, which is dictated by its bioactive compounds and their putative targets. Accordingly, this study focuses on the compounds present in the plants. Therefore, it is not essential to explore a wide range of structurally similar compounds, eradicating the need for compound structural similarity and fingerprinting.

8) Docking in general can be very subjective. It is essential to identify a viable active site and not just a protein pocket. Docking depends heavily on the size of the frame where you want to dock. Molecules will dock to any open site that is big enough which might not be the “active” site.

We have included the binding sites for the top 4 ligand-protein complexes. Please see the revisions in Section 3.9 (Line 304).

9) The paper does not mention and benchmarking studies, nor any statistical analysis on the accuracy of the predictions.

We have included positive controls in molecular docking for benchmarking. Please see the revisions in Figure 8 (Line 296) and Section 3.9. Furthermore, we put the numerical data in enrichment analyses to support Figure 6 (Line 253). Please see Tables A2-5 in Appendix A (Lines 467-474).

Reviewer 2 Report

Comments and Suggestions for Authors

This manuscript submitted by Aaron Marbyn L. Sintos et al described the Network Pharmacology reveals Curcuma aeruginosa Roxb. regulates MAPK and HIF-1 pathways to treat androgenetic alopecia”. The study presents significant insights into the therapeutic potential of C. aeruginosa in treating androgenetic alopecia (AGA) through network pharmacology and molecular docking analyses. Although the authors included good studies, a few points should be explained before accepting this manuscript, for improvement and to enhance the research and maximize its impact on related scientific society.

Comments:

  1. How specific are the identified bioactive compounds (trans-verbenol, myrtenal, carvone, alpha-atlantone, and isoaromandendrene epoxide) in targeting the key genes (CASP3, AKT1, AR, IL6, PPARG, STAT3, HIF1A, and MAPK3) involved in AGA?
  2. What are the detailed mechanisms through which each bioactive compound modulates the identified target genes and pathways (HIF-1 and MAPK)? How do these mechanisms synergize to contribute to the treatment of AGA?
  3. How do the effects of C. aeruginosa compare to current standard treatments for AGA in terms of efficacy and safety? Are there specific advantages or potential drawbacks?
  4. Beyond HIF-1 and MAPK pathways, are there other reported significant pathways modulated by C. aeruginosa in the treatment of AGA? What is the broader impact of these pathways on hair follicle health and growth?
  5. Are there any previously reported preliminary results of in vivo and in vitro studies testing the efficacy of C. aeruginosa in treating AGA? How do these results compare with the in-silico predictions? In addition, Is there any report of the dose-response relationship of the identified bioactive compounds in achieving therapeutic effects on AGA?
  6. What are the safety profiles and potential toxicities associated with the use of C. aeruginosa and its bioactive compounds in treating AGA? Are there any known side effects?
  7. How does patient variability (e.g., age, gender, genetic factors) influence the efficacy of C. aeruginosa in treating AGA? Are there specific patient populations that may benefit more from this treatment?

Exploring these questions promises to deepen the understanding of C. aeruginosa's therapeutic potential, validate its efficacy through experimental studies, and pave the way for developing innovative treatments for androgenetic alopecia. Thus, the author can make the necessary changes, and with this revision, this manuscript could be accepted for publication.

Author Response

  1. How specific are the identified bioactive compounds (trans-verbenol, myrtenal, carvone, alpha-atlantone, and isoaromandendrene epoxide) in targeting the key genes (CASP3, AKT1, AR, IL6, PPARG, STAT3, HIF1A, and MAPK3) involved in AGA?

In our study, we have demonstrated that these compounds exhibited stable binding with the hub genes by binding to active sites presented in Table 2 (Line 302), suggesting potential interactions. The implication is that the bioactive compound may modulate the activity of these genes to regulate pathways necessary to exert anti-AGA effects. However, further investigation (i.e., in vitro and in vivo studies) is required to confirm their specificity.

  1. What are the detailed mechanisms through which each bioactive compound modulates the identified target genes and pathways (HIF-1 and MAPK)? How do these mechanisms synergize to contribute to the treatment of AGA?

Currently, no studies are available to explain how each bioactive compound specifically modulates the identified target genes and MAPK and HIF-1 pathways. However, our enrichment analyses identified 3 hub genes—MAPK3, CASP3, and AKT1—implicated in both pathways through which core compounds may act on. The combined effects of these compounds through their regulation of MAPK and HIF-1 pathways, together with direct interaction with other hub genes, create a synergistic approach to address various aspects of AGA pathogenesis. Collectively, MAPK pathway promotes hair follicle proliferation, differentiation, and self-renewal to maintain hair growth cycle; HIF-1 pathway improves hair vascularization and nutrient supply which is conducive to hair growth; and AR inhibition addresses the hormonal basis of AGA.

  1. How do the effects of C. aeruginosa compare to current standard treatments for AGA in terms of efficacy and safety? Are there specific advantages or potential drawbacks?

Current research suggested that C. aeruginosa’s germacrone is on par with finasteride, exhibiting anti-androgenic activity by SRD5A inhibition to a similar extent in in vivo and vitro studies [11]. Also, clinical trials showed that the combination of minoxidil and C. aeruginosa stimulated hair growth more effectively than minoxidil alone [12,13]. In terms of safety, no side effects were reported on the use of C. aeruginosa, as opposed to those associated with finasteride and minoxidil, namely, sexual dysfunction and dermatological problems. These underscore the potential of C. aeruginosa as an AGA treatment, warranting further investigation.

  1. Beyond HIF-1 and MAPK pathways, are there other reported significant pathways modulated by C. aeruginosa in the treatment of AGA? What is the broader impact of these pathways on hair follicle health and growth?

A prior study suggested that the potential mechanism of C. aeruginosa for its anti-androgenic effect is SRD5A inhibition, which is the main action of mechanism of the present AGA drug finasteride. In relation to AGA, inhibiting SRD5A blocks the conversion of testosterone into dihydrotestosterone (DHT). Production of DHT results in the expression of growth inhibition factors that suppress hair growth, and the shortening of anagen phase that caused hair cycle imbalance, which are responsible for AGA. Thus, synergizing this with MAPK and HIF-1 pathways may influence both anti-androgenic activities and hair growth effects, promoting healthy hair follicles and eventually preventing AGA.

  1. Are there any previously reported preliminary results of in vivo and in vitro studies testing the efficacy of C. aeruginosa in treating AGA? How do these results compare with the in-silico predictions? In addition, Is there any report of the dose-response relationship of the identified bioactive compounds in achieving therapeutic effects on AGA?

As of now, there was a study conducted on the anti-androgenic effects of selected sesquiterpenes of C. aeruginosa [11]. Results showed that they inhibited SRD5A because it suppressed the growth of testosterone-induced growth of human prostate cancer cells (in vitro) and hamster flank gland model (in vivo), thereby suggesting C. aeruginosa as a novel SRD5A inhibitor. In our study, one module from the PPI cluster analysis contained SRD5A1 and SRD5A2. This suggested that C. aeruginosa interacted with SRD5A, confirming the former study. Moreover, clinical trials showed that topical application of C. aeruginosa complimented minoxidil as it increased hair growth and decreased hair shedding by enhancing penetration [12,13]. This may be caused by C. aeruginosa’s potential regulation of HIF-1 pathway which is said to increase vasodilation, promoting conducive conditions for hair growth.

There are currently no available studies that specifically report the dose-response relationship of the identified bioactive compounds from C. aeruginosa in achieving therapeutic effects on AGA. However, the findings from our study could serve as a foundation for future research to explore and establish the optimal dosages and therapeutic efficacy of these compounds in AGA treatment.

  1. What are the safety profiles and potential toxicities associated with the use of C. aeruginosa and its bioactive compounds in treating AGA? Are there any known side effects?

  1. aeruginosa has been used in traditional medicine with no reported adverse effects. Also, clinical trials involving its topical application for AGA therapy have not reported significant side effects. Further toxicological studies are recommended to establish its safety profile.

  1. How does patient variability (e.g., age, gender, genetic factors) influence the efficacy of C. aeruginosa in treating AGA? Are there specific patient populations that may benefit more from this treatment?

While AGA influences both men and women, men are more commonly and severely affected because of innate higher levels of androgens. Although C. aeruginosa exhibited trichogenic activities that could impact women too, men with genetic predispositions to AGA are likely to see more significant benefits from C. aeruginosa¸ given its anti-androgenic effects. Additionally, younger men, whose AGA symptoms are not yet more pronounced, may experience greater advantages from early intervention using it to potentially prevent AGA. Further research is recommended to fully comprehend these effects across different patient populations.

Exploring these questions promises to deepen the understanding of C. aeruginosa's therapeutic potential, validate its efficacy through experimental studies, and pave the way for developing innovative treatments for androgenetic alopecia. Thus, the author can make the necessary changes, and with this revision, this manuscript could be accepted for publication.

Round 2

Reviewer 1 Report

Comments and Suggestions for Authors

The authors have addressed the comments to my satisfaction. The paper can be published.